# Influence of SARS-CoV-2 mRNA Vaccine Booster among Cancer Patients on Active Treatment Previously Immunized with Inactivated versus mRNA Vaccines: A Prospective Cohort Study

**DOI:** 10.3390/vaccines11071193

**Published:** 2023-07-03

**Authors:** Sebastián Mondaca, Benjamín Walbaum, Nicole Le Corre, Marcela Ferrés, Alejandro Valdés, Constanza Martínez-Valdebenito, Cinthya Ruiz-Tagle, Patricia Macanas-Pirard, Patricio Ross, Betzabé Cisternas, Patricia Pérez, Olivia Cabrera, Valentina Cerda, Ivana Ormazábal, Aldo Barrera, María E. Prado, María I. Venegas, Silvia Palma, Richard Broekhuizen, Alexis M. Kalergis, Susan M. Bueno, Manuel A. Espinoza, M. Elvira Balcells, Bruno Nervi

**Affiliations:** 1Departamento de Hematología y Oncología, Escuela de Medicina, Pontificia Universidad Católica de Chile, Santiago 8330077, Chile; 2Instituto de Cáncer, Red de Salud UC-Christus, Santiago 8330032, Chile; 3Laboratorio de Infectología y Virología Molecular, Red de Salud UC Christus, Santiago 8330024, Chile; 4Departamento de Enfermedades Infecciosas e Inmunología Pediátrica, Escuela de Medicina, Pontificia Universidad Católica de Chile, Santiago 8330077, Chile; 5Departamento de Enfermedades Infecciosas del Adulto, Escuela de Medicina, Pontificia Universidad Católica de Chile, Santiago 8330077, Chile; 6Center for Cancer Prevention and Control, CECAN, Escuela de Medicina, Pontificia Universidad Católica de Chile, Santiago 8330077, Chile; 7Escuela de Medicina, Pontificia Universidad Católica de Chile, Santiago 8330077, Chile; 8Millennium Institute on Immunology and Immunotherapy, Departamento de Genética Molecular y Microbiología, Facultad de Ciencias Biológicas, Pontificia Universidad Católica de Chile, Santiago 8320000, Chile; 9Departamento de Endocrinología, Escuela de Medicina, Pontificia Universidad Católica de Chile, Santiago 8330032, Chile; 10Departamento de Salud Pública, Escuela de Medicina, Pontificia Universidad Católica de Chile, Santiago 8330032, Chile

**Keywords:** SARS-CoV-2, COVID-19, vaccine, CoronaVac, BNT162b2, cancer, chemotherapy, vaccine immunogenicity

## Abstract

**Simple Summary:**

Cancer patients receiving chemotherapy treatment are at high risk of contracting severe coronavirus disease 2019, which is associated high morbidity and mortality. Recent studies have shown that cancer patients elicit lower humoral and cellular immune responses to both inactivated vaccines and mRNA severe acute respiratory syndrome coronavirus 2 (SARS-CoV-2) vaccines. We report the results of assessing the humoral and cellular immune responses induced by the BNT162b2 vaccine booster among cancer patients receiving chemotherapy that had previously completed a primary immunization schedule with either inactivated (CoronaVac) or BNT162b2 SARS-CoV-2 vaccines. Our study demonstrated that booster vaccines elicit strong humoral and cellular responses among cancer patients receiving chemotherapy treatment, regardless of the type of vaccine used as a priming dose. No significant differences in immune response between cancer patients who were given two initial doses of either CoronaVac or BNT162b2 were detected. After adjustment for relevant covariates, the homologous regimen was associated with higher neutralizing antibody positivity and total antibody levels.

**Abstract:**

Cancer patients on chemotherapy have a lower immune response to SARS-CoV-2 vaccines. Therefore, through a prospective cohort study of patients with solid tumors receiving chemotherapy, we aimed to determine the immunogenicity of an mRNA vaccine booster (BNT162b2) among patients previously immunized with an inactivated (CoronaVac) or homologous (BNT162b2) SARS-CoV-2 vaccine. The primary outcome was the proportion of patients with anti-SARS-CoV-2 neutralizing antibody (NAb) seropositivity at 8–12 weeks post-booster. The secondary end points included IgG antibody (TAb) seropositivity and specific T-cell responses. A total of 109 patients were included. Eighty-four (77%) had heterologous vaccine schedules (two doses of CoronaVac followed by the BNT162b2 booster) and twenty-five had (23%) homologous vaccine schedules (three doses of BNT162b2). IgG antibody positivity for the homologous and heterologous regimen were 100% and 96% (*p* = 0.338), whereas NAb positivity reached 100% and 92% (*p* = 0.13), respectively. Absolute NAb positivity and Tab levels were associated with the homologous schedule (with a beta coefficient of 0.26 with *p* = 0.027 and a geometric mean ratio 1.41 with *p* = 0.044, respectively). Both the homologous and heterologous vaccine regimens elicited a strong humoral and cellular response after the BNT162b2 booster. The homologous regimen was associated with higher NAb positivity and Tab levels after adjusting for relevant covariates.

## 1. Introduction

The Coronavirus Disease 2019 (COVID-19) pandemic has caused over 6 million deaths globally [1,2]. Cancer patients receiving chemotherapy treatment represent an especially high-risk group, with an increased risk of intensive care unit admission and a higher risk of death than non-cancer patients [3,4,5]. Within the cancer patient population, multiple risk factors for poor COVID-19 outcomes have been validated, including being male, over 65 years old, a smoker, having a higher number of medical comorbidities, a high Eastern Cooperative Oncology Group performance status, and the presence of hematologic malignancies or lung cancer [6,7,8]. Consequently, worldwide efforts have led to the development, manufacture, and rapid approval of a growing number of SARS-CoV-2 vaccines with very high efficacy. The novel mRNA vaccines e BNT162b2 (Pfizer-BioNTech) and mRNA-1273 (Moderna) have been extensively distributed, with growing data showing that they provide over 94% efficacy against COVID-19 infection. These efforts have resulted in the prevention of more than 14 million COVID-19 deaths during the first year of COVID-19 vaccination, amounting to an estimated 63% reduction in total deaths [9]. However, global access has not been equitable, with many low- and middle-income countries with limited availability of mRNA vaccines opting for the emergency use of traditional, inactivated SARS-CoV2-2 vaccines such as CoronaVac (Sinovac, Beijing, China), BBIBP-CorV (Sinopharm, Beijing, China), and BBV152 (Bharat Biotech, Hyderabad, India). There is limited evidence comparing the efficacy of different vaccine regimens. A study by Hulme et al. showed that among over 300,000 health and social care workers vaccinated with BNT162b2 or ChAdOx1, the incidence of SARS-CoV-2 infection was similar [10]. Emerging studies have shown that cancer patients elicit lower humoral and cellular immune responses to both inactivated SARS-CoV-2 vaccines and mRNA vaccines [11,12,13]. In Chile, more than 70% of the population received CoronaVac as a primary vaccination. We previously reported that affliction with various immunocompromising conditions, including solid cancer, during chemotherapy markedly reduces the humoral response to two doses of the CoronaVac vaccine with neutralizing antibody (NAb) positivity, where the median neutralizing activity values were 83.1% and 51.2% for the control group versus 43.3% and 21.4% for the cancer patient group [14]. Several studies have shown that mRNA vaccines are more immunogenic and provide longer seropositivity [15]. In Chile, given that most people initially received two doses of the inactivated CoronaVac vaccine, the national vaccination program recommended a third booster dose with BNT162b2.

In this prospective observational study, we report the results of both the humoral and cellular immune responses induced by the BNT162b2 vaccine booster among solid tumor patients already receiving chemotherapy and that had previously completed, according to local guidelines, a primary immunization schedule with either inactivated (CoronaVac) or BNT162b2 SARS-CoV-2 vaccines.

## 2. Methods

We conducted an observational study to assess the comparative effectiveness of two vaccination schemes administered to immunocompromised patients in Chile in terms of their immune response. Consecutive patients with solid tumors receiving chemotherapy at Red de Salud UC CHRISTUS in Chile were invited to participate between 1 October 2021 and 28 February 2022. The study was approved by the institutional review board of the Pontificia Universidad Católica de Chile. Informed consent was obtained from all subjects involved in the study prior to their enrollment. This trial has been registered at ClinicalTrials.gov (NCT05119738) (accessed on 16 March 2023). 

### 2.1. Patients

Starting in February 2021 and according to local guidelines, all adult immunocompromised patients in Chile, including cancer patients, were offered a primary SARS-CoV-2 vaccine series. Depending on national availability, either two doses of BNT162b2, produced by Pfizer-BioNTech (3 weeks apart each), or two doses of CoronaVac, produced by Sinovac Biotech (4 weeks apart), were indicated. In addition, and following international recommendations, for all previously vaccinated patients, starting from August 2021 onwards, a booster dose with BNT162b2 vaccine was indicated. Local guidelines also recommended a period of at least 6 months between the administration of the first vaccine dose and the booster. Other inclusion criteria included age > 18 years and at least one dose of cytotoxic chemotherapy treatment prior to the third booster dose.

Cancer patients who had a previous clinical SARS-CoV-2 infection, a history of having tested positive for SARS-CoV-2, or received plasma or intravenous immunoglobulin therapy in the previous 60 days were excluded from the study. Blood samples were strictly collected 8 to 12 weeks after the administration of the BNT162b2 booster. Participants receiving initial vaccine schedule with two doses of CoronaVac followed by the BNT162b2 booster were defined as the “heterologous vaccine group”, while participants receiving a primary vaccine series with two doses of BNT162b2 followed by the BNT162b2 booster were classified as the “homologous vaccine group”.

### 2.2. Determination of Anti-SARS-CoV-2 IgG Antibodies

A commercial ELISA (SARS-CoV-2 QuantiVac, Euroimmun, Lübeck, Germany) was used for the quantitative in vitro determination of human IgG antibodies against the S1 domain of SARS-CoV-2 in serum samples in a 1:101 dilution. Data were expressed in relative units per mL (RU/mL). According to the manufacturer’s instructions, values ≥ 11 RU/mL were defined as positive. All the assays were performed in duplicate.

### 2.3. Determination of Neutralizing Antibodies against SARS-CoV-2

To determine the presence of NAb against SARS-CoV-2, we used a SARS-CoV-2 Surrogate Virus Neutralization Test (sVNT) Kit (GenScript, Piscataway, NJ, USA) in accordance with the manufacturer’s instructions. The test assesses the presence/absence of NAb and permits the interpretation of the inhibition rate as Inhibition = [1 − (OD value of Sample/OD value of Negative Control)] × 100%. As a result, a percentage of neutralization ≥ 30 at a 1:10 sample dilution is considered positive.

### 2.4. T-Cell Immune Response

Peripheral blood mononuclear cells (PBMC) were isolated from among a subset of 38 individuals using the SepMate PBMC isolation system (STEMCELL Technologies Inc., Vancouver, BC, Canada) and cryopreserved. The specific anti-SARS-CoV-2 T-cell response was evaluated using a commercial Interferon-gamma/Interleukin-2 (IFN-γ/IL-2) double-color ELISPOT assay (ImmunoSpot, Cleveland, OH, USA). Thus, T cells were stimulated using peptide megapools (MPs) derived from the SARS-CoV-2 proteome, which includes two sets of 15-mer peptides derived from the spike protein (MP-S) and the remaining proteins (MP-R), and two sets of 8- to 9-mer peptides derived from the whole SARS-CoV-2 proteome (CD8A and CD8B) [16]. A total number of 3 × 105 cells were used in each condition, positive controls were stimulated with phytohemagglutinin (PHA; #10576-015, Gibco), and mock media was used as a negative control; all conditions were assayed in duplicate. IFN-γ/IL-2 production was measured as indicated by the manufacturer, and spot-forming T cells (SFCs) were counted on an ImmunoSpot^®^ S6 Micro Analyzer (ImmunoSpot, Cleveland, OH, USA). Background spots (negative control) were subtracted from the SFC obtained for each MP stimulation and expressed as SFC per 3 × 10^5^ cells. Given that we were studying an immunocompromised population, subjects unresponsive to mitogen (positive control) were also included in the analysis.

### 2.5. Statistical Analyses 

The primary endpoint of this study was NAb seropositivity assessed 8 to 12 weeks after receiving the BNT162b2 booster vaccine. Other endpoints included the presence of SARS-CoV-2 spike protein total IgG antibodies (TAb) at a previously defined level of ≥11 RU/mL, the percentage of neutralizing activity expressed as the inhibition percentage of NAb, anti-S1 IgG geometric mean concentration (GMC), and specific T-cell immune response to SARS-CoV-2 antigens. The sample size was estimated based on the assumption that solid tumor patients receiving active treatment who received three doses of BNT162b2 would achieve 68% post-vaccine neutralizing antibody seropositivity compared to that of 43% among patients who received two doses of CoronaVac and one dose of BNT162b2, which is an assumption supported by previous studies and internal data [11,14]. We estimated that a sample size of 61 patients in each group would be sufficient to demonstrate this difference, for which a significance level of 5% (two-sided) and a statistical power of 80% were incorporated. Fisher’s exact test was used to compare categorical variables, and the Wilcoxon–Mann–Whitney test was used for continuous variables.

A multivariate analysis was performed to determine the percentages of NAb and TAb as the dependent variables. Potential confounding variables were chosen based on their statistical significance (*p* > 0.05) following a stepwise method and included the following: sex, age, lymphocyte count measured upon administration of a booster dose, hypertension, diabetes mellitus 2, asthma or chronic obstructive pulmonary disease (COPD), and chronic liver disease. The Nab percentage of inhibition was analyzed by modeling the log-transformed dependent variable using a generalized linear model specification with the identity link function and the Gaussian family function. TAb was analyzed using generalized linear models with the Gaussian family and identity link function to express the rate between geometric means through their exponentiated coefficients. 

## 3. Results

### 3.1. Patients

Between 1 October 2021 and 28 February 2022, 260 patients were invited to participate in this study, and 111 signed the informed consent form. Subsequently, two patients were excluded because they did not meet the inclusion/exclusion criteria (Figure 1). The median age of this cohort was 59.9 (IQR 48.7–65.7), and most of the patients were female (52%). The most frequent malignancy reported was colorectal cancer (50%), followed by breast cancer (16%) (Table 1).

In this cohort, 84 (77%) individuals received a primary immunization schedule involving the administration of the CoronaVac vaccine, whereas 25 (23%) received the BNT162b2 vaccine; 41 (38%) patients had started chemotherapy before receiving their first vaccine dose. The median from the BNT162b2 booster and blood sampling was 10.9 weeks. The patients in the homologous vaccine group were younger (median age 52.6 vs. 62.4 years, *p* = 0.01) and had a lower incidence of diabetes than the heterologous vaccine group (4% vs. 22%, *p* = 0.04). The median lymphocyte blood count/µL was similar in the homologous and the heterologous groups (1260 vs. 1410 *p* = 0.69). Only two patients in the heterologous vaccine group (2%) developed a symptomatic COVID-19 infection during the six-month follow-up period. 

### 3.2. Humoral Response

The humoral response of the entire cohort was high, with total IgG seropositivity and NAb positivity equal to 97% and 93%, respectively. The proportions of patients with positive total IgG antibodies for the homologous and heterologous schedule were 100% and 96% (*p* = 0.338), and the corresponding GMC titers were 173 and 158 (*p* = 0.14), respectively (Figure 2A,B).

The proportions of patients who presented NAb positivity for the homologous and heterologous regimens were 100% and 92%, respectively (*p* = 0.13). When neutralizing activity was analyzed, both groups reached a median of 100% (*p* = 0.48) (Figure 2C,D).

When the effect of relevant clinical variables on immune response was assessed in a multivariate analysis, it was observed that TAb levels were positively associated with the homologous vaccine scheme (geometric mean ratio of 1.41; *p* = 0.044) and with a higher lymphocyte count (geometric mean ratio of 1.00025; *p* = 0.006). As described in the Methods section, the effect of NAb on the participants’ log-transformed variables was examined; hence, the beta coefficients corresponded to semi-elasticities. The results showed a statistically significant effect of the homologous vaccine scheme, accounting for an expected 26% increase in the inhibition percentage of neutralizing antibodies, in relative terms, (*p* = 0.027). The lymphocyte counts also showed a significant effect, with an expected increase of 0.01% in the inhibition percentage of neutralizing antibodies for each increased unit of the lymphocyte count (*p* = 0.005). No other variables were associated with both NAb and TAb seropositivity (Table 2).

### 3.3. T-Cell Response

A subgroup of 38 patients in our cohort was evaluated with respect to T-cell response upon stimulation with MP of SARS-CoV-2-derived peptides. The IFN-y response in the homologous vaccine group (*n* = 16) when stimulated with 15-mer peptides (MP-S+MP-R) or 8- to 9-mer peptides (CD8A+CD8B) showed a response similar to the heterologous regimen (*n* = 22), with a median of 27.5 versus 27.25 SCF/3 × 10^5^ cells for MP S+R (*p* = 0.53) and 80.5 versus 44.5 SCF/3 × 10^5^ cells (*p* = 0.27) for CD8A+B, respectively (Figure 3A). Likewise, no difference was observed in the IL-2 response upon stimulation, with a median of 31.3 versus 28.5 SCF/3 × 10^5^ cells for MP S+R (*p* = 0.65) and 21 versus 16 SCF/3 × 10^5^ cells for CD8A+B (*p* = 0.71) (Figure 3B), respectively.

## 4. Discussion

The purpose of this study was to estimate the comparative effectiveness of two different vaccination schemes applied to immunocompromised patients in Chile in terms of their immune capacity. Our findings indicate that both the homologous and heterologous vaccine schedules elicited a strong humoral response, which was determined by measuring total IgG antibodies and neutralizing antibody seropositivity 8 to 12 weeks after receiving the BNT162b2 booster. While the results showed no statistical differences between the seropositivity of NAb in either group (defined as ≥30%) nor with respect to TAb (≥11 UR/mL), when we explored the effect of potential confounding variables through multivariate analysis, we found a significant statistical difference favoring the homologous group, which was consistent for both the percentage of NAb and Tab levels. These results might be relevant in the long term when protection time is taken into consideration. A multivariate analysis using seropositivity, our primary endpoint, was not possible because in the homologous group, all 25 patients surpassed the predefined positivity level. Furthermore, in the subgroup analysis of patients who had T-cell responses measured using both IFN-y and IL-2 levels after stimulation with SARS-CoV-2-derived peptides, no differences were noted between the two schedules. 

There is strong and growing evidence that for immunocompromised patients, including solid tumor cancer patients receiving chemotherapy treatment, SARS-CoV-2 vaccines elicit a weaker serologic response compared to that elicited in the normal population. Consequently, having an increased risk of breakthrough infections leads to worse outcomes overall when compared to non-cancer patients, with reports showing an up to 27% higher risk of hospitalization with a 5% increase in death rate [14,17]. A systematic review and meta-analysis including 19 studies investigating vaccine immunogenicity among patients with solid tumors undergoing chemotherapy showed a poorer response to a COVID-19 vaccine versus those not on active treatment [18]. A growing body of evidence suggests a strong correlation between neutralizing antibody titers and protection against SARS-CoV-2 variants of concern that is effected through a reduction in symptomatic infection and severe disease risk [19]. However, vaccine effectiveness wanes over time, with some accounts reporting efficacy falling below 50% after the first year following vaccination [20]. This phenomenon seems to be particularly important among older adults, as reported in a Brazilian study where a waning of the level of protection against severe outcomes was observed for individuals aged ≥80 years compared to younger patients at 120 days or more after the booster dose [21]. These results reinforce the importance of booster vaccination policies, particularly with respect to high-risk populations. Our results suggest that the pragmatic strategy used in multiple developing countries of combining inactivated and mRNA-based COVID-19 vaccines is effective in ensuring humoral and cellular immune responses against SARS-CoV-2 among high-risk groups. These results are consistent with a recent systematic review that concluded that both homologous and heterologous vaccination regimens achieve high humoral immune responses against the Omicron variant, particularly when a booster dose of mRNA vaccine is included [22,23].

Most studies evaluating immune responses among cancer patients have included a wide range of patients, incorporating hematological patients who are known to be exposed to deeper immunosuppressive treatments or solid tumor patients treated with immunotherapy [24,25]. Not all cancer treatments have the same effect on the immune response. In a recent cohort study, cancer patients treated with checkpoint inhibitors had similar humoral and cellular immune responses to mRNA COVID-19 vaccines compared to healthy donors, whereas cancer patients receiving B-cell-directed therapies had a much lower response [26]. In this study, we included a more homogeneous population of solid tumor patients, among whom 76% had colorectal, breast, or gastric cancer and were treated only with moderate immunosuppressive cytotoxic chemotherapy. However, the question remains determining which one is the best approach. Our work showed no differences between homologous and heterologous schedules in terms of both humoral and cellular immune responses; however, a longer follow-up will allow us to determine if there are clinical differences among our oncological population regarding booster doses and if, considering the growing number of SARS-CoV-2 variants, heterologous vaccination schedules show any benefit [27,28]. Recent evidence indicates that different SARS-CoV-2 variants of interest show differential reductions in neutralization and replication by antibodies elicited by COVID-19 mRNA vaccines [29]. In this uncertain setting, multiple countries, including Chile, have recommended a fifth SARS-CoV-2 BNT162b2 vaccine, with some initial data showing significant and rapid increases in antibody titers after the fourth BNT162b2 dose [30].

Our work has presented several limitations. First, the design of the study precluded the analysis of clinical outcomes given the limited sample size and short follow-up period. Hence, we are not able to confirm how this high level of immune response translates into fewer hospitalizations or deaths in these patients. Second, we did not include different SARS-CoV-2 variants of interest in our analysis. Third, there are some imbalances in the baseline characteristics and the sample size of the homologous and heterologous vaccine groups that are associated with the lack of randomization due to the observational nature of our study. Therefore, we performed a multivariate analysis to adjust for potential confounding variables.

## 5. Conclusions

This observational study shows that booster vaccines elicit strong humoral and cellular responses in cancer patients receiving cytotoxic treatment regardless of the type of vaccine used as a priming dose. No significant differences in immune response between cancer patients who were given two initial doses of either CoronaVac or BNT162b2 were detected. After adjustment for relevant covariates, the homologous regimen was associated with higher NAb positivity and TAb levels.

## Figures and Tables

**Figure 1 vaccines-11-01193-f001:**
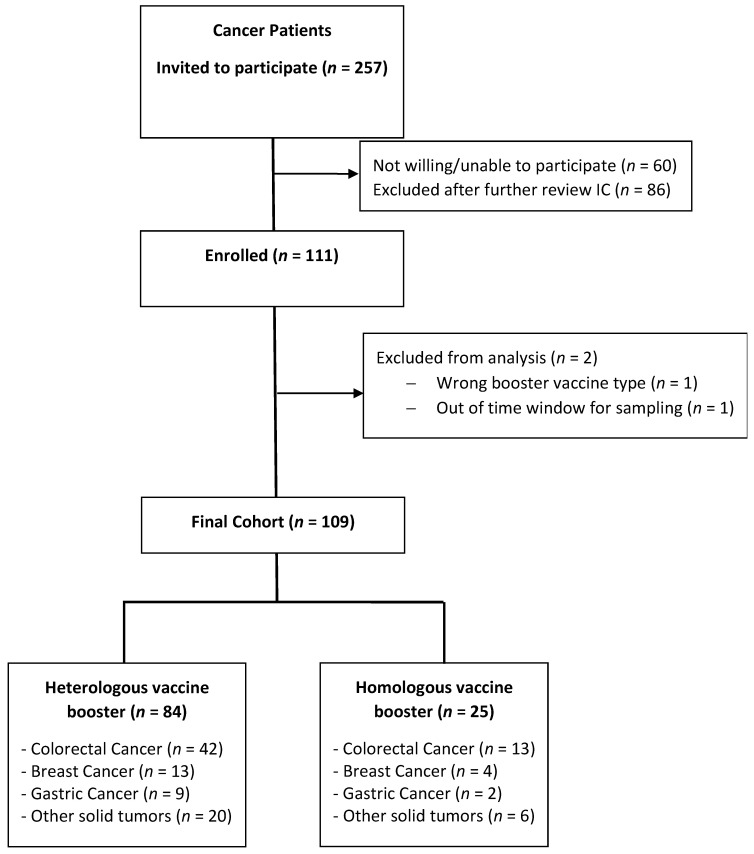
Study flow chart.

**Figure 2 vaccines-11-01193-f002:**
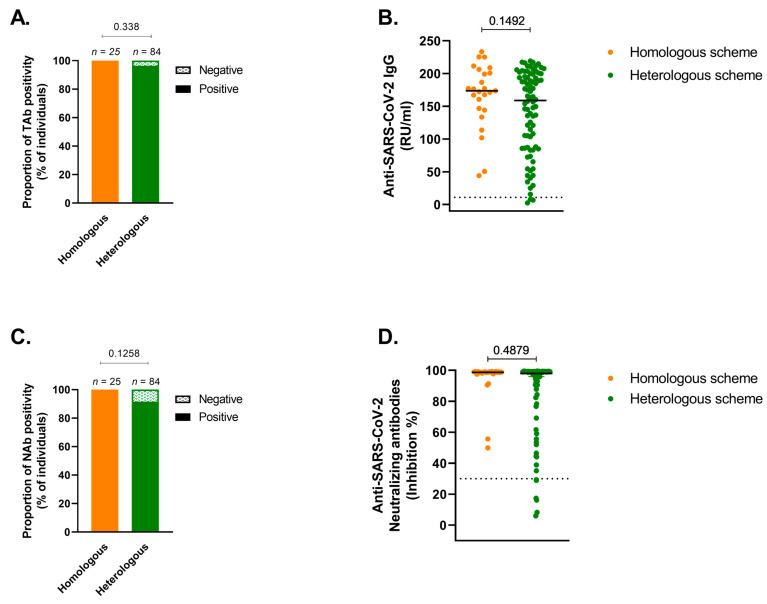
Humoral response against SARS-CoV-2 among solid cancer patients receiving either a homologous or heterologous vaccination schedule. (**A**) proportion of total IgG (TAb) anti-S1 positivity (≥11 relative units per mL, RU/mL), (**B**) total IgG anti-S1 GMC (95%CI), RU/mL), (**C**) proportion of neutralizing antibody (NAb) positivity (≥30% of inhibition rate), and (**D**) neutralizing activity (median (IQR) of percentage of inhibition). Dotted lines in (**B**,**D**) indicate seropositivity cut-offs.

**Figure 3 vaccines-11-01193-f003:**
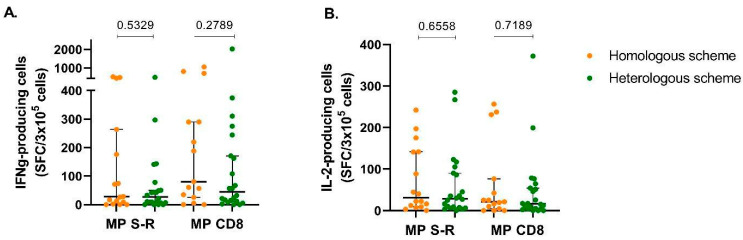
Evaluation of IFN-ɣ- and IL-2-secreting Spot-Forming T cells in patients with either homologous or heterologous vaccination schedules. PBMCs (3 × 10^5^ cells) in both the homologous group (*n* = 17) and the heterologous group (*n* = 23) were stimulated with a peptide megapool (MP S-R) or a peptide megapool (MP CD8) from SARS-CoV-2 proteins. (**A**) IFN-γ-secreting spot-forming T cells (SFC) and (**B**) IL-2-secreting SFC were quantified using ELISPOT. Medians and 95% CI are shown.

**Table 1 vaccines-11-01193-t001:** Patients’ clinical characteristics.

	Homologous Vaccine Scheme (*n* = 25)	Heterologous Vaccine Scheme (*n* = 84)	*p* Value
Median Age (range)	52 (10)	58 (13)	0.051
Sex (%) Female Male	12 (48) 13 (52)	45 (53) 39 (45)	0.147
Comorbidities (%) Hypertension Diabetes Mellitus Asthma/COPD Chronic Liver disease	5 (20) 1 (4) 0 0	30 (36) 19 (22) 4 (5) 3 (4)	0.22 0.04 0.57 1.00
Diagnoses (%) Colorectal Cancer Breast Cancer Gastric Cancer Other solid tumors	13 (52) 4 (16) 2 (8) 6 (24)	42 (50) 13 (15) 9 (10) 20 (24)	0.457
Stage Localized Metastatic	8 (32) 17 (68)	29 (35) 55 (65)	0.815
Median Lymphocyte blood count (IQR)	1260 (1005 –1855)	1410 (1020 –1910)	0.69
Chemotherapy before first COVID19 vaccination Yes No	9 (36) 16 (64)	32 (38) 52 (62)	0.849

**Table 2 vaccines-11-01193-t002:** Summary of multivariate analysis that includes vaccine groups and lymphocyte counts.

	Nab ^a^		Tab ^b^	
Variable	Coef. (95% CI)	*p* Value	GMR (95% CI)	*p* Value
Vaccine group				
Heterologous group	Reference		Reference	
Homologous group	0.26 (0.029–0.50)	0.027	1.41 (1.009–1.989)	0.044
Lymphocyte count	0.0001 (0.00005–0.0002)	0.005	1.00025 (1.0000–1.0004)	0.006

^a^ Percentage of SARS-CoV-2-neutralizing antibodies (Nab); ^b^ Total anti-SARS-CoV-2 S1 IgG antibodies (Tab) (UR/mL); Abbreviations: GMR—geometric mean ratio; CI—confidence interval; Coef.—beta coefficient obtained from the generalized linear model, representing the semi-elasticity of the covariate with respect to the log transformation of the response-dependent variable.

## Data Availability

No new data were created or analyzed in this study. Data sharing is not applicable to this article.

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
