# Peer review of "Influence of SARS-CoV-2 mRNA Vaccine Booster among Cancer Patients on Active Treatment Previously Immunized with Inactivated versus mRNA Vaccines: A Prospective Cohort Study"

_vaccines, 2023, doi:10.3390/vaccines11071193_

Round 1
Reviewer 1 Report
The article is devoted to an important area of research - the study of the immunogenicity of vaccines in cancer patients. The article presents data on the formation of humoral and cellular immune response after vaccination in patients with various types of tumors.
As I got acquainted with the work, a number of comments arose:
1. In the abstract, the authors write that their goal was to determine the efficacy, however, the article provides data only on immunogenicity. Need to be corrected.
2. It is not clear from the text of the article when the vaccination with BNT162b2 was carried out after the primary vaccination.
3. It would be interesting and useful to present data on volunteers with no history of cancer. This will allow comparing the level of immune response recorded after vaccination.
Author Response
REVIEWER 1:
Comments and Suggestions for Authors
The article is devoted to an important area of research - the study of the immunogenicity of vaccines in cancer patients. The article presents data on the formation of humoral and cellular immune response after vaccination in patients with various types of tumors.
As I got acquainted with the work, several comments arose:
- In the abstract, the authors write that their goal was to determine the efficacy, however, the article provides data only on immunogenicity. Need to be corrected.
Thank the reviewer for these comments. Our goal was to determine efficacy, but as you point out this is not equivalent to immunogenicity. We have modified this as requested by the reviewer.
- It is not clear from the text of the article when the vaccination with BNT162b2 was carried out after the primary vaccination.
We agree this is not clear, and we have made the corresponding changes to make sure it is well established. We defined that vaccination boost with BNT162B2 should have been given 8 to 12 weeks prior to data recollections. Please see the attached file for the revised manuscript.
- It would be interesting and useful to present data on volunteers with no history of cancer. This will allow comparing the level of immune response recorded after vaccination.
We thank the reviewer for this comment. In a previous publication in our group (Balcells et al. Clin Infect Dis. 2022 Aug 24;75(1):e594-e602), we compared the immune response in 5 cohorts of immunocompromised patients with a group of 67 controls after 2 doses of CoronaVac. NAb positivity and median neutralizing activity were 83.1% and 51.2% for the control group versus 43.3% (P < .001) and 21.4% (P<.01 or P = .001) in the cancer with solid tumors group. While these results provide some clarification to the comment raised by the reviewer, we agree this is still a limitation of our work since the current work includes patients who also received the BNT162b2 booster. We plan on presenting a longer follow up to determine booster dose efficacy where we could compare with non-cancer volunteers.

Reviewer 2 Report
The aim is stated clear. The authors stated clearly what study found and how they did it. The title is informative and relevant. Appropriate and key studies are included.
The process of selection of the subjects was clear. The variables are well defined and measured appropriately. The study methods are valid and reliable. There are enough details provided in order to replicate the study.
The data is presented in an appropriate way. Results are discussed from different angles and placed into context without being overinterpreted.
The conclusions answer the aim of the study. The conclusions are supported by references and own results. The limitations of the study are not fatal, but they are opportunities to inform future research.
Specific comments on weaknesses of the article and what could be improved:
Major points
Minor points
1. Some of abbreviations are not introduced in the summery.
Author Response
REVIEWER 2:
Comments and Suggestions for Authors:
The aim is stated clear. The authors stated clearly what study found and how they did it. The title is informative and relevant. Appropriate and key studies are included.
The process of selection of the subjects was clear. The variables are well defined and measured appropriately. The study methods are valid and reliable. There are enough details provided in order to replicate the study.
The data is presented in an appropriate way. Results are discussed from different angles and placed into context without being overinterpreted.
The conclusions answer the aim of the study. The conclusions are supported by references and own results. The limitations of the study are not fatal, but they are opportunities to inform future research.
Specific comments on weaknesses of the article and what could be improved:
Minor points
- Some of abbreviations are not introduced in the summary:
We agree with the reviewer and have made the changes accordingly, please see attached the revised manuscript.

Reviewer 3 Report
Review of the manuscript entitled SARS-CoV-2 mRNA Vaccine Booster in Cancer Patients on Active Treatment Previously Immunized with Inactivated Versus mRNA Vaccines: A Prospective Cohort Study.
The authors investigated the humoral and cellular immune response to both inactivated vaccines and mRNA BNT162b2 SARS CoV-2 vaccines. They report the results assessing the humoral and cellular immune responses induced by the BNT162b2 vaccine booster in cancer patients receiving chemotherapy having previously completed a primary immunization schedule with either inactivated (CoronaVac) or BNT162b2 SARS-CoV-2 vaccines. Their study demonstrated that booster vaccines provide high humoral and cellular response in cancer patients receiving chemotherapy treatment regardless of the type of vaccine used as a priming dose.
Comments and Suggestions for Authors
I think it is a very thorough study about the immunization against COVID19-SARS-CoV-2 in cancer patients. However it should be submitted to a more specific journal into some immunology journal.
Author Response
REVIEWER 3:
Comments and Suggestions for Authors:
Review of the manuscript entitled SARS-CoV-2 mRNA Vaccine Booster in Cancer Patients on Active Treatment Previously Immunized with Inactivated Versus mRNA Vaccines: A Prospective Cohort Study.
The authors investigated the humoral and cellular immune response to both inactivated vaccines and mRNA BNT162b2 SARS CoV-2 vaccines. They report the results assessing the humoral and cellular immune responses induced by the BNT162b2 vaccine booster in cancer patients receiving chemotherapy having previously completed a primary immunization schedule with either inactivated (CoronaVac) or BNT162b2 SARS-CoV-2 vaccines. Their study demonstrated that booster vaccines provide high humoral and cellular response in cancer patients receiving chemotherapy treatment regardless of the type of vaccine used as a priming dose.
Comments and Suggestions for Authors
- I think it is a very thorough study about the immunization against COVID19-SARS-CoV-2 in cancer patients. However, it should be submitted to a more specific journal into some immunology journal.
We appreciate the comment of the reviewer who recommends that we send this manuscript to a more specific journal, such as a journal dedicated to immunology. However, we strongly believe that our findings showing the impact of different COVID vaccination strategies in patients receiving chemotherapy for cancer are of interest for the audience of the prestigious journal Cancers. Cancer is the first or second cause of death in different countries of the world, and the COVID pandemic seriously impacted treatment opportunities.
